# *Escherichia coli* Biofilm Formation, Motion and Protein Patterns on Hyaluronic Acid and Polydimethylsiloxane Depend on Surface Stiffness

**DOI:** 10.3390/jfb13040237

**Published:** 2022-11-11

**Authors:** Annabelle Vigué, Dominique Vautier, Amad Kaytoue, Bernard Senger, Youri Arntz, Vincent Ball, Amine Ben Mlouka, Varvara Gribova, Samar Hajjar-Garreau, Julie Hardouin, Thierry Jouenne, Philippe Lavalle, Lydie Ploux

**Affiliations:** 1INSERM UMR-S 1121 Biomaterial Bioengineering, Centre de Recherche en Biomédecine de Strasbourg, 67084 Strasbourg, France; 2Faculty of Dentistry, University of Strasbourg, 67000 Strasbourg, France; 3PISSARO Proteomic Facility, IRIB, 76130 Mont-Saint-Aignan, France; 4Mulhouse Materials Science Institute, CNRS/Haute Alsace University, 68057 Mulhouse, France; 5Polymers, Biopolymers, Surfaces Laboratory, CNRS/UNIROUEN/INSA Rouen, Normandie University, 76821 Rouen, France; 6CNRS, 67037 Strasbourg, France

**Keywords:** surface stiffness, hydration, polydimethylsiloxane (PDMS), hyaluronic acid, biofilm, bacterial mobility, protein patterns

## Abstract

The surface stiffness of the microenvironment is a mechanical signal regulating biofilm growth without the risks associated with the use of bioactive agents. However, the mechanisms determining the expansion or prevention of biofilm growth on soft and stiff substrates are largely unknown. To answer this question, we used PDMS (polydimethylsiloxane, 9–574 kPa) and HA (hyaluronic acid gels, 44 Pa–2 kPa) differing in their hydration. We showed that the softest HA inhibited Escherichia coli biofilm growth, while the stiffest PDMS activated it. The bacterial mechanical environment significantly regulated the MscS mechanosensitive channel in higher abundance on the least colonized HA-44Pa, while Type-1 pili (FimA) showed regulation in higher abundance on the most colonized PDMS-9kPa. Type-1 pili regulated the free motion (the capacity of bacteria to move far from their initial position) necessary for biofilm growth independent of the substrate surface stiffness. In contrast, the total length travelled by the bacteria (diffusion coefficient) varied positively with the surface stiffness but not with the biofilm growth. The softest, hydrated HA, the least colonized surface, revealed the least diffusive and the least free-moving bacteria. Finally, this shows that customizing the surface elasticity and hydration, together, is an efficient means of affecting the bacteria’s mobility and attachment to the surface and thus designing biomedical surfaces to prevent biofilm growth.

## 1. Introduction

The contamination of medical surfaces by bacteria is inextricably linked to many infections in clinical practice, leading to the increased use of antibiotics and the subsequent emergence of antibiotic-resistant bacteria [1,2]. These causal bacteria are also known to form difficult-to-remove biofilms on medical surfaces; thus, the need to develop materials that inhibit bacterial colonization on medical surfaces is of the utmost priority [3]. In biofilm formation, pioneer bacteria attach to the surface of materials, form colonies, and produce a protective polymer matrix composed of polysaccharides and other biomolecules. Biofilms are typically highly difficult to eradicate, primarily because of the limited drug diffusion in their extracellular matrix and the physiological states (resistance and dormancy) that these bacteria can switch between to protect themselves [4].

Surface mechanical properties emerged some ten years ago as a potential tool for controlling the colonization of biomedical materials by bacteria [5,6,7,8]. Indeed, bacteria can sense the mechanical properties of their environment and quickly and appropriately respond to them [9,10]. The stiffness of a host’s extracellular matrix [11] or a material’s surface in contact with the bacteria modulate the interactions between the bacteria and their environment [5,6,7,8].

There are two uncertainties limiting the exploitation of surface mechanical properties to prevent biofilm formation on medical devices. First, recent studies on polydimethysiloxane (PDMS), polyelectrolyte multilayers (PEMs), poly(ethylene glycol) dimethacrylate (PEGDMA) and agar hydrogel surfaces showed that the influence of stiffness on bacterial adhesion differs according to the substrate. The stiffer the PDMS materials are [12,13], or the softer hydrogels or PEMs are, the less bacteria are adhered [6,7,8,14,15]. The reason for this marked difference is still unclear, as these observations could be due to the presence or absence of the culture medium used, variations in the bacterial species, and significant differences in the material hydration (typically observed between PDMS and agar hydrogels). Secondly, the biological actors and pathways regulated in this type of material–bacteria interface are still unknown, but they are necessary to elucidate the bacterial mechano-sensing and underlying mechanisms in the expansion or prevention of bacterial populations on soft and stiff surfaces.

In this study, we determined the behavior and proteome of *Escherichia coli* (*E. coli*) placed in contact with both hydrated and non-hydrated surfaces of differing elasticities. The short- and long-term colonization (i.e., biofilm formation) were both considered, as well as the capacity of the bacterial population to expand on the surface through individual bacterial mobility. We conducted this study on PDMS and on hyaluronic-acid-based (HA) materials. PDMS elastomers are non-degradable synthetic polymers commonly used to design medical devices and implants, such as breast implants, contact lenses, or catheters, while HA hydrogels have been used more recently as biomaterials for dermal wound repair and as scaffolds in tissue engineering [16,17]. The HA and PDMS materials were designed to have Young’s moduli ranging from around 10 Pa to a few kPa and from a few kPa to hundreds of kPa, respectively. The stiffest HA and the softest PDMS materials were adapted to provide similar Young’s moduli (a few kPa) to enable a direct comparison between these materials. The material’s ability to store aqueous liquids (through water content measurements) was determined as an index of the hydration capacity. The materials were also characterized in terms of their surface chemistry, hydrophobicity/hydrophilicity, topography, and elasticity to describe the surface parameters that directly impact bacterial adhesion. The adhesion, retention (i.e., the fraction of adhered bacteria retained after the creation of an air–surface interface), and mobility of the sessile bacteria, as well as the further biofilm formation, were evaluated in situ by fluorescence confocal microscopy during the first and longer period of colonization on the hydrated and non-hydrated materials of differing surface elasticities. The protein patterns of the adhered bacterial cells were determined using a quantitative proteomic approach.

## 2. Materials and Methods

### 2.1. Synthesizing the PDMS Materials

The PDMS materials were prepared using the SYLGARD™ 184 Silicone Elastomer Kit (Dow Corning Corporation, Midland, MI, USA). The material’s stiffness was adjusted by varying the curing agent-to-base mass ratio from 1:80 (“PDMS-9kPa”) to 1:5 (“PDMS-574kPa”). For each ratio, the elastomer base and curing agent were thoroughly mixed. The mixture was poured into a plastic cup, cured at 120 °C for 20 min, and incubated at room temperature for 24 h to ensure complete polymerization. The polymerized elastomer was cut into 8 mm-diameter disks using a circle cutter previously sterilized under UV for 30 min. The synthesized PDMS materials were stored at room temperature until further use. The samples had a thickness ranging from 1.5 mm to 2.0 mm (measured with a caliper). The chemical structure of the PDMS polymer is illustrated in Figure 1A, and images of the samples are shown in the Appendix A.

### 2.2. Synthesizing the HA Hydrogel Materials

The hyaluronic acid (HA) hydrogels were created using the 1,4-butanediol diglycidyl ether (BDDE) crosslinker, as described in a previous work [18]. Briefly, a 0.038 mM solution of HA (Lifecore Biomedical, Chaska, MN, USA; MW = 823 kDa) in NaOH (0.25 M) was mixed with 10% and 30% BDDE (*v*/*v*) to prepare the “HA-44Pa” and “HA-2kPa” materials, respectively. The mixture was poured into a 35 mm-diameter Petri dish and allowed to crosslink at 37 °C for 72 h. The HA hydrogel was further cut into 8 mm disks using a circle cutter previously sterilized with UV for 30 min. The HA hydrogel samples were stored at 4 °C until further use. The samples were about 2.0 mm thick under room conditions. The chemical structure of the HA hydrogel polymer is illustrated in Figure 1B, and images of the samples are shown in the Appendix A.

### 2.3. Characterizing the Surface Chemistry

The chemical composition of the topmost sample surfaces was determined using X-ray photoelectron spectroscopy (XPS) on a VG SCIENTA SES-2002 spectrometer equipped with an Al Kα monochromatic X-ray source (1486 eV) at a power of 420 W. XPS analysis was carried out under a pressure of 10^−9^ mbar on areas of about 4 mm × 6 mm with a pass energy of 500 eV for the survey spectra and 100 eV for the high-resolution spectra of carbon (C1s), oxygen (O1s), and silicon (Si2p). The peaks were fitted by Gaussian–Lorentzian functions using XPS-CASA software (casaXPS software 2.3.18 Ltd., Teignmouth, UK) after subtracting the Shirley background [19]. The atomic concentration analysis was based on the integrated peak area of each component and took into account the sensitivity factor, mean free pathway of an electron, and the transmission function of the analyzer. The binding energies were selected based on the information in the literature, and all the components were defined in reference to the CHx/C-C component at 285.0 eV [20,21].

### 2.4. Characterizing the Surface Topography

The surface topography of the PDMS and HA materials was analyzed using atomic force microscopy (AFM). AFM images were acquired in the contact mode with a Bioscope AFM microscope (Veeco, Santa Barbara, CA, USA) and a ScanAsyst^®^ Fluid cantilever probe (Bruker AFM Probes, Billerica, MA, USA) in silicon nitride with a reflective gold-back side coating (spring constant: 0.7 N/m; triangular geometry; tip radius: 20 nm). The analysis was performed at a resolution of 512 × 512 pixels at a scan rate of 0.5 Hz. The material samples were immersed in phosphate-buffered saline solution (PBS) during the analysis. One 20 µm × 20 µm zone was acquired per sample. AFM images were acquired, and the mean surface roughness (*Sa*) was calculated using Gwyddion^®^ software [22].

### 2.5. Characterizing the Surface Wettability

The surface wettability was determined by measuring the water contact angles in the static mode. The measurements were performed using an Attension^®^ Theta tensiometer (Biolin Scientific, Västra Frölunda, Sweden). The sessile drop method was used to evaluate the contact angle at equilibrium (Ɵ) of a 5 µL distilled drop of water on the sample surface. Briefly, the drop was deposited onto the material surface via a 0.7 mm-inner-diameter needle, and 1 min-long videos were recorded using a CCD camera (1 image/0.07 s) from the moment when the water droplet hit the material until it reached a stable shape, i.e., when the equilibrium was reached. The contact angle was evaluated at this point. Values of Ɵ were extracted from the images using numerical fits of the droplet shape based on the Young–Laplace model [23]. The results obtained are presented as the mean ± standard deviation of at least two measurements per material. The material surfaces were defined as either hydrophobic (when Ɵ > 90°) or hydrophilic (when Ɵ < 90°) [24]. For the HA hydrogels, the precise measurement of the contact angles was not possible due to the fast absorption of the water drop by the material.

### 2.6. Characterizing the Material Hydration

The hydration of the PDMS and HA materials was evaluated by measuring their water content after full hydration. The material samples were weighed before immersion (*w*_1_) in distilled water for 24 h at room temperature. After this period, the excess water was removed very delicately with absorbent paper, and the samples were weighed again (*w*_2_). They were then dehydrated in an oven at 30 °C for 72 h and weighed (*w*_3_) for a third time. The drying time *t*_0_ was chosen as the minimal duration needed to measure constant *w*_3_ values for *t* > *t*_0_. The drying was thus considered to be complete at this time *t*_0_. The water content was calculated according to Equation (1). The measurements were repeated on three independent samples for each material. The material hydration results are presented as the mean ± standard deviation of the three measurements.
(1)$$Water\;content\;\left(\%\right)=100\times \frac{w_{2}-w_{3}}{w_{2}}$$

### 2.7. Characterizing the Surface Elasticity

The elasticity of the material’s surface was studied using a Chiaro^®^ nano-indenter (Optics11 Life, Amsterdam, The Netherlands). Nanoindentation analysis was performed in the liquid phase (M63 medium [25]) according to the ferrule top indentation method [26]. The indentation of a spherical glass probe in the tested material was calculated through interferometric detection. The spring constant of the cantilever and the radius of the spherical probe were selected to match the surface properties (2.90 N/m and 23.5 µm for PDMS-574kPa; 0.53 N/m and 25.5 µm for PDMS-9kPa; 0.53 N/m and 25.5 µm for HA-2kPa; 0.027 N/m and 24.5 µm for HA-44Pa). The measurements were performed in a grid pattern, with 20 µm between the two successive indentation locations. The approach velocity was fixed at 5 µm s^−1^. The measurements were repeated 25 times, 29 times, 28 times, and 25 times for the PDMS-574kPa, PDMS-9kPa, HA-2kPa, and HA-44Pa materials, respectively. Young’s moduli (*E*) were calculated using the Hertz model [27]. The results are shown as the mean ± standard deviation.

### 2.8. Bacterial Strains and Growth Medium

Microbiological experiments were conducted using the *E. coli* K-12 SCC1 strain [28], (purchased by Prof. Chun Chau Sze, Nanyang Technological University, Singapore), which constitutively expresses the green fluorescent protein (GFP), produces colonic acid and flagella, and forms biofilms. Bacteria from a −80 °C frozen stock were grown overnight at 30 °C on lysogeny broth (LB) (Difco™, Tucker, GA, USA) agar plates. Then, a colony was suspended in LB medium and incubated at 30 °C overnight. A total of 3 mL of the first culture was then added to 27 mL of fresh LB. After 3 h of incubation at 30 °C, the suspension was centrifuged at 3000 rpm for 20 min, and the bacterial pellet was re-suspended in M63G medium [25] for the experiments with live bacteria. For the experiments with dead bacteria, the bacterial pellet was re-suspended in a solution of 3.7% (*w*/*v*) paraformaldehyde in PBS. The suspension was homogenized by vortexing and agitation for 1 min and 3 h, respectively, at room temperature. The suspension was then centrifuged at 3000 rpm for 20 min and resuspended in M63G. A total of 100 µL of this suspension was spread on an LB agar Petri dish and incubated at 30 °C for 24 h to verify the bacterial cell death. The optical density of the live or dead bacteria suspensions (i.e., absorbance at 600 nm) was adjusted to 0.01, corresponding to approximatively 5 × 10^6^ bacteria/mL.

### 2.9. Bacterial Culture and Sample Preparation for Adhesion, Retention, and Mobility Analyses

The material samples were set on a glass coverslip with a dot of biocompatible glue, placed in a 12 mm-diameter home-made sample-holder (Appendix A), and immersed with 1.5 mL of *E. coli* K-12 SCC1 suspension before incubation at 30 °C for 3 h. Glass coverslips were used as the internal controls in each experiment. For the adhesion and mobility analyses, the planktonic bacteria were removed using a washing procedure based on dilutions in order to avoid any de-wetting of the sample surface and the related risk of driving away the sessile bacteria due to the action force applied by the triple line [29]. A total of 500 µL of the bacterial planktonic suspension was therefore carefully removed and replaced with 500 µL of fresh M63 medium (i.e., M63G without glucose). This was repeated three times to reach an OD value of about 0 of the removed supernatant. For the analysis of the retention (i.e., the fraction of adhered bacteria retained after the creation of an air-surface interface), the washing procedure was performed as follows: all the supernatant (1.5 mL) was removed without contact with the material, and 500 mL of fresh M63 medium was then delicately added and removed. This procedure was repeated three times.

### 2.10. Bacterial Culture and Sample Preparation for Biofilm (Long-Term Colonization) Analysis

The samples were prepared as described above for the adhesion, retention, and mobility analysis, but a longer culture time was employed (72 h). Briefly, the material samples, having been set on a glass coverslip with a dot of biocompatible glue and placed in a 12 mm-diameter home-made sample-holder, were immersed with 1.5 mL of *E. coli* K-12 SCC1 suspension before incubation at 30 °C for 3 h. Planktonic bacteria were then removed using the same washing procedure as that used for the adhesion analysis. The samples were further incubated under static conditions for the remaining time (69 h).

### 2.11. Microscopy Analysis

The adhesion, retention, and mobility of the sessile bacteria, as well as the biofilms, were analyzed with a confocal laser scanning microscope (CSLM) (LSM710, Zeiss, Oberkochen, Germany) equipped with a 50× objective lens (Objective LD EC Epiplan-Neofluar 50×/0.55 DIC M27; working distance = 9.1 mm). A 488 nm excitation wavelength laser was used to excite the GFP produced by *E. coli* K-12 SCC1 and to enable the detection of the bacterial cells. The fluorescence emitted was thus collected from 493 nm to 578 nm for the adhesion, retention, and mobility analyses, or from 493 nm to 544 nm for the biofilm analyses. For the biofilm analyses, the collected range of the wavelength was reduced to enable the staining of the biofilm matrix with another fluorescence dye (Texas Red™ concanavalin A; see below). For the adhesion and retention analyses, at least 5 images for each sample were taken in situ in the final washing medium. For the mobility analyses, at least 5 one-minute videos (1 image every 3, 4 or 5 s, depending on the experiment) were taken in the same conditions. For the biofilm analyses, the biofilm matrix was previously stained with Texas Red™ concanavalin A (ConA) (Invitrogen™, Waltham, MA, USA), which selectively binds to α-mannopyranosyl and α-glucopyranosyl residues. The ConA powder was diluted in a 0.1 M sodium bicarbonate (NaHCO_3_) solution to obtain a 2 mg/mL concentration, which was added to the final washing medium. The final ConA concentration was 125 µg/L. After 30 min of incubation at 30 °C in the dark, at least 5 3D images (Z-stacks) were taken in situ in the final washing medium. The 561 nm excitation wavelength laser was used to excite the ConA. The fluorescence emitted by ConA was collected between 582 nm and 650 nm. The locations analyzed were determined randomly for all the analyses. Each experiment was repeated at least 3 times. The number of adhered and retained bacterial cells was determined from the microscopy images through the automatic counting of the individual cells using ImageJ^®^ 2.0 software [30] and the Analyze Particles plugin. The cell tracking of the sessile bacteria in the one-minute videos was performed using the NIS-Elements Advanced Research Imaging software^®^ (Nikon Group™, Tokyo, Japan) with the automated object tracking module. A list of the consecutive coordinates for the position of each bacterium during the one-minute video was generated. The further analysis of the bacterial mobility was performed using these data. The biovolume per surface unit corresponding to the cell and matrix of the biofilm, that is, the GFP- and ConA-related biovolumes per surface unit, respectively, were extracted using the Comstat2^®^ software and expressed in µm^3^/µm^2^ [31].

### 2.12. Data and Statistical Analysis

The statistical significance of the differences in the adhered or retained bacteria numbers, as well as the biofilm features (biomass), was determined using *t*-tests. The H_0_ hypothesis (*µ*_1_ = *µ*_2_) was rejected for *p*-values < 0.05, *p*-values < 0.01, or *p*-values < 0.001 depending on the experiment (see figure captions). The raw data used for the mobility analysis were the *E. coli* (x, y) positions at successive times. Only the paths that could be followed for over approximately 1 min (or, precisely, 55, 57, or 76 s, depending on the experiment) were selected. The analysis focused on the values of *d_net_*, which is the net distance travelled by a bacterial cell from the first image (at *t* = 0 of the experiment) to the image observed at a given time *t* during the experiment (Figure 2). The experimental cumulative relative frequency F of *d_net_*^2^ of each sample was determined for each observation time.

If all the bacteria had free Brownian motion with the same diffusion coefficient *D*, the complementary cumulative relative frequency, Φ = 1 – F, would be a decreasing exponential function of *d_net_*^2^, i.e., $\Phi =\exp\left(-\frac{d_{net}^{2}}{4Dt}\right)$. In other words, as a general rule, when Φ is known, Φ(*x*_0_) represents the probability of finding a value of *x* ≥ *x*_0_. In our observations, however, Φ was not a decreasing exponential function of *d_net_*^2^. We thus devised a model that assumed that the bacteria sampled in each experiment encompassed *m* subpopulations with homogeneous characteristics of motion. In each subpopulation, bacteria were considered as moving in a harmonic energy well, *U*(*r*) (where *r* is the position of the bacterium in reference to the well center), with a characteristic radius (*r_C_*) defined by *U*(*r_C_*)/*k*_B_*T* = 1, where *k*_B_*T* is the thermal energy (Equation (2)). The movement is Brownian, with a diffusion coefficient *D*, though not exclusively, due to thermal agitation. At a radial distance *r* from the center of the well, the bacterium experiences a restoring force, as modeled by a Hooke spring. In these conditions, preliminary simulations made it possible to determine the dependence of <*d_net_*^2^> on the parameters *D* and *r*_C_ and on the time *t* (Equation (3)). This equation is similar to that identified by Meijering et al. [32], because <*d_net_*^2^> is the mean squared displacement (MSD) taken at the final observation time. The free Brownian motion is recovered when *r_C_* tends toward infinity (Equation (4a)), whereas <*d_net_*^2^> tends toward a limit imposed by the well when *t* tends toward infinity (Equation (4b)):(2)$$\frac{U\left(r\right)}{k_{B}T}={\left(\frac{r}{r_{C}}\right)}^{2}\;$$
(3)$$\langle d_{net}{}^{2}\rangle =2\;r_{C}{}^{2}\;\left(1-e^{-2Dt/r_{C}{}^{2}}\right)$$
with
(4a)$$\underset{r_{C}\to \infty }{\lim}\langle d_{net}{}^{2}\rangle =4\;D\;t$$
and
(4b)$$\underset{t\to \infty }{\lim}\langle d_{net}{}^{2}\rangle =2\;r_{C}{}^{2}$$

It was then possible to calculate the function Φ of *d_net_* for the *m* subpopulations, according to Equation (5):(5)$$\Phi =\sum_{i=1}^{m}w_{i}\;e^{-d_{net}{}^{2}/{\langle d_{net}{}^{2}\rangle }_{i}}$$
where <*d_net_^2^*>_i_ depends on the (*D_i_*, *r_C,i_*) couple, and *w_i_* is the relative weight of the *i*th subpopulation (the weights are subject to the constraints $\sum_{i=1}^{m}w_{i}=1$ and *w_i_* ≥ 0 for any *i*).

The same preliminary simulations also enabled us to determine the expression of the mean total length, <*d*_tot_>, of the trajectories, following Equation (6):(6)$$\langle d_{tot}\rangle =\sqrt{\frac{\pi }{2}}\frac{t}{\Delta t}r_{C}\;\sqrt{1-e^{-2D\Delta t/r_{C}{}^{2}}}$$
where *t* is the observation duration and Δ*t* is the time lag between the successive pictures.

The mean speed can then be obtained merely by dividing <*d*_tot_> by the corresponding time *t*, according to Equation (7):(7)$$\langle v\rangle =\sqrt{\frac{\pi }{2}}\frac{1}{\Delta t}r_{C}\;\sqrt{1-e^{-2D\Delta t/r_{C}{}^{2}}}$$

In the special case when *r*_C_ ⟶ ∞, <*v*> reduces to Equation (8):(8)$$\langle v\rangle =\sqrt{\frac{\pi D}{\Delta t}}$$

The challenge was then to identify the *m* triplets (*D_i_*, *r_C,i_*, *w_i_*). To this end, an optimization method known as “simulated annealing” was used. The aim was to reduce, as much as possible, the sum (*S)* of the squares of the differences between the experimental Φ_exp_ and the computed one, Φ_comp_. For a fixed number of subpopulations, *m*, and an initial starting “temperature”, *T*_ini_, the weight *w_i_* corresponding to (*D_i_*, *r_C,i_*) of each of the *m* triplets is varied to reduce *S*. After a number of minimization trials, the temperature is lowered (e.g., by a factor of 1.05) until no new configuration can be accepted according to the Metropolis criterion [33]. The process goes on until no further reduction in E can be achieved. The quantity E, which plays the role of the energy, and the quantity called “temperature” are borrowed from the simulated annealing method first developed in statistical physics. The pre-processing of the experimental data and simulated annealing were performed with home-made computer codes written in Fortran^®^. From the triplet (*D_i_*, *r*_*C,i*_, *w_i_*), a confinement index (*ρ_i_* ∈ [0, 1]) could be defined using Equation (9):(9)$$\rho_{i}=\frac{{\langle d_{net}{}^{2}\rangle }_{i}}{2\;r_{C,i}{}^{2}}$$
for each of the *m* subpopulations.

### 2.13. Analysis of the Whole Proteome: Protein Extraction

Proteomic analyses of the cells adhered to the HA-44Pa and HA-2kPa and those adhered to the PDMS-9kPa and PDMS-574kPa materials were carried out. The HA and PDMS substrates were inoculated for 3 h and rinsed as described above. The adhered cells were delicately detached with a cell scraper. The scraped cells were removed from the cell scraper through fresh M63 rinses. The solution containing the cells was transferred to 1.5 mL Eppendorf tubes, which were centrifuged at 10,000 rpm for 30 min. The supernatant was then removed, and the pellet was stored at −20 °C until the time of analysis. The proteins were extracted by two freezing cycles and sonication in a lysis buffer (7 M urea, 2 M thiourea, 4% 3-[(3-cholamidopropyl) dimethylammonio]-1-propanesulfonate hydrate (CHAPS), 65 mM dithiothreitol (DTT), 25 mM Tris/HCl). The protein concentrations were determined using the Bradford assay [34].

### 2.14. Enzymatic Digestion

Twenty micrograms of protein were mixed with SDS loading buffer (63 mm Tris-HCl, pH 6.8, 10 mm DTT, 2% SDS, 0.02% bromphenol blue, 10% glycerol) and loaded onto an SDS-PAGE stacking gel (7%). A short electrophoresis procedure was performed (10 mA, 45 min and 20 mA, 2 h) to concentrate the proteins. After migration, the gels were stained with Coomassie Blue and unstained (50% ethanol, 10% acetic acid, 40% deionized water). The protein band revealed from each fraction was excised, washed with water, and immersed in a reductive medium (5 mm DTT). Cysteines were irreversibly alkylated with 25 mm iodoacetamide in the dark. Following the washing steps using water, the gel bands were subjected to protein digestion with trypsin (1 μg per band) overnight at 37 °C in ammonium bicarbonate buffer (10 mm, pH 8). The peptides were extracted with H_2_O/CH_3_CN/TFA mixtures (49.5/49.5/1) and dried. For each growth condition, three biological replicates were performed, and two technical replicates were conducted for each of them (in total, 6 samples per condition were analyzed).

### 2.15. Tandem Mass Spectrometry

The peptides were analyzed using mass spectrometry. All the experiments were performed using LTQ-Orbitrap Elite equipment (Thermo Scientific, Waltham, MA, USA) coupled with an Easy nLC II system (Thermo Scientific). One microliter of sample (1 µL) was injected into an enrichment column (C18 PepMap100, Thermo Scientific). The separation was performed with an analytical column needle (NTCC-360/internal diameter: 100 µm; particle size: 5 µm; length: 153 mm, NikkyoTechnos, Tokyo, Japan). The mobile phase consisted of H_2_O/0.1% formic acid (FA) (buffer A) and CH_3_CN/FA 0.1% (buffer B). The tryptic peptides were eluted at a flow rate of 300 nL/min using a three-step linear gradient: from 2 to 40% of buffer B over 75 min, from 40 to 80% of buffer B over 4 min, and 80% of buffer B for 11 min. The mass spectrometer was operated in the positive ionization mode with the capillary voltage and the source temperature set at 1.5 kV and 275 °C, respectively. The samples were analyzed using the collision-induced dissociation (CID) method. The first scan (MS spectra) was recorded using the Orbitrap analyzer (Rs = 60,000) in the mass range of *m*/*z* 400–1800. Then, the 20 most intense ions were selected for the tandem mass spectrometry (MS^2^) experiments. Singly charged species were excluded from the MS^2^ experiments. The dynamic exclusion of the already fragmented precursor ions was carried out for 30 s with a repeat count of 1, a repeat duration of 30 s, and an exclusion mass width of ±10 ppm. Fragmentation occurred in the linear ion trap analyzer at a collision energy of 35 eV. All measurements in the Orbitrap analyzer were performed through on-the-fly internal recalibration (lock mass) at *m*/*z* 445.12002 (polydimethylcyclosiloxane).

### 2.16. Protein Quantification

A label-free experiment was performed as previously described by Obry et al. [35]. Briefly, after the MS analysis, the raw data were imported into the Progenesis LC-MS software (Nonlinear Dynamics, version 4.0.4441.29989, Newcastle, UK). For the comparison, one sample was set as a reference, and the retention times of all the other samples within the experiment were aligned. After the alignment and normalization, a statistical analysis was performed for the one-way analysis of variance (ANOVA) calculations. For the quantitation, peptide features presenting with a *p*-value and a q-value of less than 0.05 and a power of more than 0.8 were retained. The MS/MS spectra of the selected peptides were exported for peptide identification with Mascot (Matrix Science, version 2.2.04) against the database restricted to *E. coli* K12 MG1655 from NBCI. Database searches were performed with the following parameters: 1 missed trypsin cleavage site was allowed, and the variable modifications were the carbamidomethylation of cysteine and oxidation of methionine. The mass tolerances for the precursor and fragment ions were set at 5 ppm and 0.35 Da, respectively. False discovery rates (FDR) were calculated using a decoy-fusion approach in Mascot (version 2.2.04). The identified peptide–spectrum-matches with a −10log *p*-value of 20 or higher were kept at an FDR threshold of 5%. The Mascot search results were imported into Progenesis. For each condition, the total cumulative abundance of the protein was calculated by summing the abundances of the peptides. Proteins identified with less than 2 peptides were discarded. Only the proteins with a 2-fold variation in their average normalized abundances between growth conditions were retained.

## 3. Results and Discussion

The HA (from (44 ± 16) Pa to (2.2 ± 0.6) kPa)) and PDMS (from (9 ± 2) kPa to (574 ± 11) kPa) materials cover a wide range of surface elasticities, as well as opposite hydration properties (Figure 3A and Appendix A). As expected, the Young’s modulus *E* (analyzed by nanoindentation) increased with the increasing crosslinking of both materials. It is important to note that the stiffest HA (HA-2kPa) and the softest PDMS (PDMS-9kPa) revealed similar Young’s moduli. This trend was confirmed by the bulk properties evaluated by rheometry (Appendix A), although a quantitative difference was expected due to the difference in stimulation between the two methods (perpendicular to the surface or oscillating for the nanoindentation, shear stress for rheometry). The rheometry also revealed that, as expected, the PDMS with 1:40 and 1:20 curing-to-base ratios had an intermediate stiffness (10.2 ± 0.7 and 34 ± 9) (Appendix A). Furthermore, consistent with the literature [36], elastic behavior predominated in all the materials except for PDMS-9kPa, which displayed a significant viscous behavior, as previously reported by Valentin et al. [37]. However, increasing the crosslinking does not change the viscous component of HA materials, whereas it only weakly increases that of PDMS materials. It is important to note that nanoindentation, rather than rheology, can effectively detect the surface elasticity sensed by bacteria when they are in contact with a material’s surface. Therefore, in the following sections, we describe the surface elasticity using the Young’s modulus *E* obtained by nanoindentation.

As expected, the water content was low for the PDMS materials, varying from about 1% to 5%. On the contrary, the HA materials displayed typical hydrogel behavior, with water contents of approximately 97% and more than 99% for HA-2kPa and HA-44Pa, respectively. The hydrophobic or hydrophilic characters were in accordance with those expected for the material’s hydration capacity. The values of the water contact angle Ɵ at equilibrium were less than 10° on both HA surfaces (they could not be measured more accurately due to the rapid water absorption of the hydrogel materials), whereas the Ɵ values showed an expected hydrophobic nature for both PDMS-9kPa and PDMS-574kPa (115 ± 1° and 114 ± 2°, respectively) [13,38,39,40].

In addition to elasticity, the surface topography and chemistry can directly act bacterial adhesion. Herein, topography is quantified by the roughness, which provides a general view of the surface, and AFM images complete the description regarding the texture and morphology. The Sa values were less than 4 nm for all the materials (2.9 nm, 2.8 nm, 3.8 nm, and 2.5 nm for the HA-44Pa, HA-2kPa, PDMS-9kPa, and PDMS-574kPa materials, respectively), and only rare peaks were present (maximum height of approximately 30 nm and 90 nm for the HA and PDMS materials, respectively) (Figure 3B–E). Chemistry is defined by the chemical groups that may be involved in specific chemical interactions with the receptors on the bacterial surface and by the hydrophobicity/hydrophilicity, which may hamper the contact of a bacterium with the surface if a layer of water molecules is present on the material surface or enable hydrophobic interactions between the bacterial outer surface and the material surface on hydrophobic surfaces. The surface charge was not considered, since it cannot significantly impact bacterial adhesion in a culture medium of a high ionic strength (about 150 mM for LB and M63G). Electrostatic interactions between the charges carried by the bacterium and the material surface are strongly reduced in such ion-rich media, as illustrated by the Debye length, which was found to be as small as 0.8 nm [41,42].

### 3.1. E. coli Adhesion and Retention and Biofilm Growth Have an Opposite Correlation with Stiffness on the HA and PDMS Surfaces

The bacterial adhesion (i.e., measured after 3 h of culture in situ without the retrieval of the liquid environment) and retention (i.e., measured after the surface retrieval from the liquid) increased significantly as the PDMS surface elasticity decreased. Although the results for the HA surfaces suggested an opposite trend, an intermediate stiffness value between HA-44Pa and HA-2kPa would be required to conclude a monotone relationship. Through adhesion tests, we showed that the adhesion rises by a factor of 1.8 on the HA surfaces when *E* increases from 44 Pa to 2.2 kPa, whereas it rises by a factor of 1.3 on the PDMS surfaces when *E* decreased from 574 kPa to 9 kPa (Figure 4 and Appendix A). The retention followed the same trends on both the PDMS and HA materials (Figure 4A). However, cell numbers were reduced by more than 50% in comparison with the adhesion tests. This was probably caused by the shear stress applied to the sessile bacteria by the triple line of the de-wetting front created through the retrieval of the liquid during the retention test (Appendix A) [43,44,45]. After 72 h, PDMS-574kPa and HA-44Pa were less colonized than PDMS-9kPa and HA-2kPa, respectively, demonstrating that the general trend observed at 3 h was maintained (Figure 4B). However, the PDMS-9kPa and HA-2kPa materials were similarly colonized by GFP-producing cells after 72 h of culture, while significantly fewer bacteria were adhered on HA than on the PDMS materials after 3 h. The production of the polymeric matrix also depended on the material. The amount of matrix production was much higher on PDMS-9kPa than on the HA-2kPa material, and it was almost completely absent on the HA-44Pa material. These results indicate that variations in the surface elasticity and surface hydration result in variations in the biofilm and affect several key stages of its formation: (1) bacterial adhesion, (2) the growth of the cell population, and (3) the production of the polymeric matrix.

In the literature (Appendix A), the results obtained after retention assays show trends on PDMS materials similar to those in the present study [12,46]. Only a few works have considered bacterial adhesion rather than retention. They report comparable numbers of bacteria on soft and stiff PDMS [40,47] or even slightly lower numbers on the softest compared to the stiffest surfaces [48]. These differences in the reported results are most probably caused by the conditions used when washing the samples and harvesting the bacterial cells after culture on the surfaces, in addition to differing bacterial cells, culture media, cell densities for inoculation, and the methods used to count the bacteria. To our knowledge, results regarding bacterial colonization on HA hydrogels of differing elasticities have never been published. The reported data concern hydrogels formed of agarose [6,14,15] and other hydrated materials based on polyethylene glycol [15,49,50], poly(N-isopropylmethacrylamide) [49] or polyelectrolytes [7], sometimes derived from HA [8]. Apart from Wang et al. [50], all the authors concluded there was a decrease in bacterial adhesion or retention with decreasing material elasticity (Appendix A), in agreement with the present study. Wang et al. reported an opposite trend in the case of polyacrylamide hydrogels with elasticity values of less than 1 kPa [50]. However, the significant difference in the surface topography of the 17-Pa PAAm and 654-Pa PAAm materials used in their work probably impacted the retention of the bacteria on the surface. In the present study, differences in the material surface chemistry and topography were not expected to directly modulate the bacterial adhesion. Indeed, the PDMS and HA substrates had similar flat topographies in relation to the size of a bacterium, which did not significantly change with the degree of crosslinking (Figure 3B–E). According to the literature, it is fair to assume that such a topography is unlikely to cause significant inter-sample variations in bacterial adhesion and retention [51]. Furthermore, the variation in the degree of crosslinking between the soft and stiff materials did not profoundly change the surface chemistry on the extreme surfaces of either the HA or PDMS materials (Figure 1). However, the surface chemistry differed, as expected, between these two materials (Table 1 and Appendix A). Binding energies attributed to C–Si and O–Si bonds and silicon oxides (SiO_2_ or SiO_4_) were detected on the PDMS materials, whereas C–C, C–H, C–O, and C=O bonds were detected on the HA hydrogels. This difference cannot directly result in a difference in bacterial adhesion, since the chemical groups present on the surfaces of PDMS and HA are not specific ligands of the bacterial membrane [52,53]. However, it can indirectly lead to differences due to the resulting surface charge and hydrophobic character. As previously noted, the difference in the surface charge cannot significantly impact bacterial adhesion in a culture medium of a high ionic strength. However, the hydrophobic or hydrophilic property that the detected groups conferred on the surface probably had an impact on the bacterial adhesion. The values of the water contact angle Ɵ at equilibrium (less than 10° on HA surfaces and about 115° on PDMS surfaces) are consistent with a significant difference in the adhesive properties between highly hydrophilic HA and hydrophobic PDMS surfaces. Water molecules present on the HA material surface probably impede the attachment of the bacteria to the surface [54,55], whereas hydrophobic surfaces, such as PDMS materials, may be more favorable to attractive interactions between the surface and a bacterium [56,57]. Therefore, on PDMS, bacterial adhesion decreases when the stiffness increases, and this can be considered as the actual effect of the surface elasticity. On the contrary, the effect of surface hydration may offset the effect of the surface elasticity on HA materials. The hydration of HA surfaces in comparison to PDMS also probably hinders the production of the extracellular polymer matrix or its attachment to the surface. Indeed, water molecules on the hydrophilic surface may hamper the bacterial sensing of the surface and, subsequently, prevent the physiological change from a planktonic to sessile state [58]. The polymer matrix may also have insufficient tethers and, therefore, an insufficient stability to remain on such a hydrated surface [55].

Finally, the differences in the bacterial adhesion and retention on the HA and PDMS surfaces with different elasticity most probably resulted from the difference in the elastic character, which was offset the by hydrophilic character of the HA surface. These surface properties may have caused differences in the bacterial biology, particularly in regard to the surface sensing of, and tethering to, the material surface [59].

### 3.2. MscS and FimA Abundances Vary with Surface Stiffness and Surface Hydration

We next determined whether these combinations of surface stiffness and surface hydration regulate protein expression in *E. coli*. (Figure 5A). On the HA materials, MscS and 8 other proteins (Lon, RpoS, GatC, SecYEG, RplV, RpmC, RplB, RhlE; Table 2) were more abundant on the HA-44Pa surface than on the HA-2kPa surface, among 900 proteins. Five proteins were more abundant on the HA-2kPa surface than on the HA-44Pa surface (DnaK, PykF, Eno, AldA, RpsA). On the PDMS materials, FimA and 7 other proteins (AdhE, Lpp, PolA, LacZ, FtsY, Mog, TopA; Table 3) were more abundant on the PDMS-9kPa surface than on the PDMS-574kPa surface among approximately 1000 proteins. Two proteins were in higher abundance on the PDMS-574kPa surface than on the PDMS-9kPa surface (CysE and PurD). Furthermore, four proteins were found to be accumulated on both the HA-2kPa and PDMS-9kPa compared to the HA-44Pa and PDMS-574kPa surfaces, respectively (Table 4A). However, their maximum fold change (MFC) values were low, except for the AdhE protein on the PDMS surfaces. All these data point to an alteration in the protein patterns according to the material.

Specifically, one remarkable protein in high abundance on HA-44Pa was MscS, a mechanosensitive channel of a low conductance. MscS is known as a membrane tension sensor [60]. It opens in response to stretch forces in the membrane lipid bilayer and is believed to be sensitive to the mechanical deformation of the cell wall induced by surface contact. This triggers surface-specific cellular responses. On HA-44Pa, bacteria probably sense the mechanical characteristics using MscS channels, as they do in a liquid environment [10,61]. This may be possible due to the high softness and hydration of this material, which might provide adequate stimuli for the MscS channels. Furthermore, RpoS, which is overproduced on HA-44Pa, is associated with the transition to the stationary state at the beginning of the bacterial adhesion to a surface [62], and Lon protease regulates bacterial motility during this type of transition in several species [63,64]. Their overproduction on HA-44Pa suggests that the bacteria are in a very early stage of adhesion on this surface. In contrast, the higher abundance of DnaK observed on HA-2kPa is consistent with a more advanced stage of bacterial adhesion on this surface, as shown by the bacterial adhesion and biofilm formation assays (Figure 4). DnaK is associated with the production of curli [65], which are necessary for bacteria to tether themselves to surfaces and further strengthen their biofilms [66]. It should be noted that proteins in the curli appendages were not identified in the proteomes. This may have resulted from a low production of curli by the bacteria but also from the low solubility of such amyloid fibers [67], which may have led to their removal during the sample preparation prior to the mass spectrometry analysis. Thus, the protein production is consistent with the initiation or formation of biofilms on both HA surfaces, but the surface stiffness of the HA-2kPa positively regulates the production of proteins that are favorable to adhesion compared to the HA-44Pa. As a consequence, adhesion and further biofilm production are probably delayed on the HA-44Pa surface compared to the HA-2kPa surface, in agreement with the quantity of biofilm measured on these surfaces (Figure 4).

Several proteins in higher abundance on PDMS-9kPa are also involved in biological processes related to biofilm formation. Thus, the overproduction of FimA is related to the higher production of Type-1 pili, which is an appendage used by *E. coli* to attach to abiotic surfaces [68,69] (see the SEM micrographs of *E. coli* SCC1 producing pili in Appendix A) and is associated with high levels of adhesion and biofilm formation. Interestingly, Type-1 pili are also sensitive to changes in the bacterial mechanical environment [70] and allow the bacteria to attach to surfaces in a force-dependent, so-called “catch-bond” manner [71]. The production of AdhE, a multifunctional and key metabolic enzyme in bacterial physiology and pathogenicity, and Lpp, a major outer membrane protein of *E. coli*, are also positively correlated with biofilm formation [72,73]. In particular, the Lpp protein is linked to CsgA expression [73], which is the major subunit of the curli. In contrast, the higher abundance of CysE on PDMS-574kPa is consistent with the observed limitation on biofilm growth. Indeed, this enzyme has been associated with the inhibition of *E. coli* biofilm formation [74]. Thus, like HA surfaces, PDMS surfaces displayed protein production consistent with the initiation or formation of biofilms. However, the surface stiffness of the PDMS-574kPa negatively regulates the production of proteins that are favorable to adhesion compared to the PDMS 9kPa. As a consequence, the biofilm formation process probably progressed with a delay on the PDMS-574kPa surface compared to the PDMS-9kPa surface, in agreement with the quantity of biofilm measured on these surfaces (Figure 4). Furthermore, the characterization of the lower amount of FimA on the HA surfaces compared to the PDMS-574kPa surface (Table 4B) confirms that bacterial adhesion was just beginning on HA surfaces, whereas it was at a more advanced stage on the PDMS surface.

In summary, the bacterial mechanical environment significantly regulates the MscS mechanosensitive channel (with a higher abundance on the least colonized HA-44Pa) and Type-1 pili [70] (with a higher abundance on the most colonized PDMS-9kPa). Busscher and coworkers recently demonstrated that the contact of *Staphylococcus aureus* with a surface creates an adhesion force triggering the mechanosensitive channel MscL [60]. Here, the mechanosensitive channel MscS is probably triggered in *E. coli* by a similar mechanism. Because of the low critical force necessary to open these channels, it may even allow the bacterium to distinguish between elasticities that are favorable or detrimental to their survival on the surface. Furthermore, Type-1 pili allow bacteria to attach to surfaces in a force-dependent, so-called “catch-bond” manner due to changes in the pili structure (coiling, uncoiling) [76,77]. This ability of Type-1 pili to retract under external forces means that they are potential sensors of surface elasticity, as well as adaptive tethers. The structure and quantity of Type-1 pili may change depending on the surface stiffness for a given shear stress condition. A similar process may occur when an adherent bacterium is exposed to an increase and decrease in shear stress for a given surface stiffness.

Finally, the difference in the protein patterns confirms that tethers (Type-1 pili) are rare or almost absent on the hydrated HA surfaces, whereas they are far more abundant on the PDMS surfaces. This is consistent with the expected role of Type-1 pili as a tether favorable to bacterial adhesion. On PDMS-9kPa, in particular, the high abundance of Type-1 pili is expected to strengthen the link between the bacteria and the surface and, thus, to enhance the bacterial stability and further bacterial growth on the surface.

### 3.3. Bacteria Are Confined to All Surfaces except for PDMS-9kPa and Are Less Diffusive on Soft Surfaces

Next, given that tethers are understood to hinder mobility, we investigated whether mechanical conditions govern the adherent bacteria’s ability to move, which is critical for bacterial colonization. Since the proteins associated with *E. coli* motility were not in significant abundance in the protein patterns, we presumed that the mobility was merely a consequence of Brownian motion. The fraction of mobile *E. coli* cells in the total sessile population, defined as the cells that moved at least 2 µm from their initial position (*d_net_* at the maximal observation time, Figure 2), varied according to the material (Figure 6). It was similar on HA-2kPa, PDMS-9kPa, and PDMS-574kPa but was approximately 3 times lower on HA-44Pa. Along with the low quantity of sessile bacteria, most of the “immobile” bacteria observed on HA-44kPa probably revealed their difficulty in maintaining contact with this surface. Most likely, only the bacteria anchored to the surface remained sessile. In contrast, the slightly adhered bacteria, such as those moving on a surface, were easily removed by any slight movement of the liquid due to the temperature or the washing process. On the contrary, the moving bacteria adhered tightly enough to the other types of materials (HA-2kPa, PDMS-9kPa and PDMS-574kPa) to maintain contact with the surface. In addition, the indentation of the material by sessile bacteria is expected to be more than 10 times higher on a surface with a Young’s modulus of 44 Pa (1.8 nm) than on surfaces with Young’s moduli above 2 kPa, (0.13 nm for 2 kPa) (Appendix A). This may have favored frictional forces during mobility and the subsequent reduction in bacterial movements. Finally, as the mobility of bacterial cells on surfaces is understood to facilitate the expansion of the sessile population, this may have restricted the colonization of the HA-44Pa in comparison to the patterns that occurred on the other surfaces. This is in agreement with the low colonization observed in the short and long term on this surface.

It is important to note that the mobility of the bacterial cells previously killed with PFA solution, which were thus considered as inert objects, (Figure 6A) was much lower than the mobility of the live cells, regardless of the material (from 62% to 98% less). This suggests that the characteristics of the live cells were, to a significant extent, crucial for the observed mobility. These characteristics may have been purely biological, enabling, for example, active mobility or active anchorage on the surface, or they may have been related to the physicochemical properties of the bacterial surface and its appendages. However, they could have been modified by the PFA treatment.

Furthermore, the motion of the live bacteria significantly differed between cells on the same material surface, as shown by the frequency of *d_net_* for each material type (typical examples in Figure 6B–E). More specifically, the non-linearity of logΦ with respect to *d_net_*^2^ (measured at a given time on a given sample) suggests that several subpopulations of the bacteria formed the whole population (example of the Φ function of a PDMS-574kPa sample in Figure 7A; please refer to the part on “Data and statistical analysis” in the Experimental section for more details). Such a coexistence of bacterial subpopulations differing in their mobility is not surprising and is known to occur in planktonic and sessile populations [78,79]. However, only Song et al. reported that such a heterogeneity may vary according to the surface stiffness [80]. In addition, the relationship of < *d_net_*^2^> with *t* suggested that, on some surfaces, the bacteria were confined to one area rather than diffusing freely. Indeed, this relationship was linear with the statistical fluctuations on several surfaces, as expected for a free Brownian motion, but also showed a strong deviation from this law on other surfaces (examples in Appendix A). This limitation in the cell movement may correspond to their attachment to the surface through an elastic bond.

To identify whether the sessile population might vary in these terms depending on the surface nature and surface stiffness, we modeled the Φ functions with the *m* subpopulations of the bacterial cells with homogeneous motion characteristics, and we considered the bacterial cells as Brownian particles moving in harmonic energy wells (Equation (2)). It is worth noting that the aim here was to dissect the observation rather than to propose a final explanatory model. Therefore, we chose to consider Brownian motion with different levels of confinement depending on the parameter *r_C_* for the sake of simplicity, but it is possible that, in reality, confined and sub-diffusion Brownian motions may be combined [32]. Starting with the hypothesis that the population encompasses several subgroups, the modeling of the 19 experimental Φ functions of a PDMS-574kPa sample (after times 3, 6, …, 57 s) with *m* = 50 subpopulations confirmed that the total population was accounted for. Each of these *m* subpopulations was defined by *D*, the diffusion coefficient, *r_C_*, the radius, and *w*, the relative fraction in the whole population (Figure 7B), as well as by the confinement index *ρ* as the quantification of the level of freedom of the bacteria within a subpopulation (Equation (7) and Figure 7C,D). According to convention, the bacteria were considered free in the case of *ρ* ≤ 0.2 and confined in the case of *ρ* ≥ 0.8. This example shows that fewer than 10 subpopulations account for more than 99% of the tracks (Figure 7B). This suggested that we ought to restrict the number of subpopulations to *m* = 10 for the further modeling. The typical results of the modeling with *m* = 10 that best fit the experimental Φ functions of the <*d_net_*^2^> measured on a sample are shown in Appendix A for each material type. The results confirm that the tracks (each corresponding to one bacterium) can be grouped into few subpopulations (from 3 to 8, depending on the sample). Only rare bacteria (less than 1% of the total tracked population) were identified with other motion characteristics.

In general, the results show that the diffusion of the cells increased globally with the surface stiffness. In other words, the subpopulations with higher diffusion coefficients were more frequent or contained more cells on the materials with a higher stiffness (Figure 7E–G), whereas, on the contrary, subpopulations with low diffusion coefficients were more frequent when the surface stiffness decreased. In this respect, the subpopulations on the PDMS-574kPa surface revealed diffusion coefficients higher than those of the subpopulations identified on the PDMS-9kPa (Figure 7F), and the diffusion coefficients on surfaces of similar elasticities (HA-2kPa and PDMS-9kPa) were distributed similarly (Figure 7G). In addition, the bacterial population on the HA-44Pa surface was composed of subpopulations with diffusion coefficients that usually were smaller (maximum at 10^−3^ µm^2^ s^−1^) than those of the subpopulations on the HA-2kPa surface (maximum at 10^−2^ µm^2^ s^−1^) (Figure 7E). Furthermore, the confinement of the cells did not globally increase or decrease with the surface stiffness. Rather, the fraction of free bacteria (i.e., with *ρ* from 0 to 0.2) increased with bacterial adhesion, though without significant correlation (R^2^ = 0.78; Figure 8). Typically, the bacterial subpopulations were the most confined on HA-44Pa, the least colonized surface. They were less confined on the HA-2kPa and PDMS-574kPa materials (Figure 7H) and revealed a predominant non-confined motion on PDMS-9kPa.

Overall, these results indicate that diffusion and confinement are not correlated, but that diffusion correlates with the surface stiffness, whereas confinement oppositely correlates with the adhesion rate. Typically, bacteria tend to move very little on HA-44Pa, while they move slightly more on HA-2kPa and much more on PDMS-574kPa but stay at a nearly constant location on the three surfaces. On the PDMS-9kPa surface, the bacteria tend to move as little as they do on HA-2kPa surface but move much further away from their initial location (Figure 5B). This displacement, characterized as being free on the basis of our model, is associated with greater bacterial adhesion, greater biofilm formation, and a significantly higher abundance of Type-1 pili compared to the other surfaces (Figure 5A). This suggests that free motion is crucial for allowing bacteria to colonize a surface. These observations are consistent with the significant contribution to population expansion attributed to bacterial mobility by other authors [81]. However, free displacement may be in contradiction with the higher abundance of Type-1 pili on the PDMS-9kPa surface. Indeed, pili are thought to be responsible for the tethering of bacterial cells to a surface and for stable bacterial adhesion [82].

Nevertheless, pili may also be favorable to bacterial mobility on surfaces, as suggested by the works of Busscher’s and Vogel’s groups [60,83,84]. In this respect, Sjollema et al. demonstrated that the successive detachment and attachment of multiple tethers between the bacteria and the surface, such as those created by pili, can lead to sub-micrometric displacements under thermal agitation [60,83,84]. Thomas et al. also reported the combination of rolling motion and the pili attachment of *E. coli* cells on a surface under various flow rates [83]. Such a type of motion may be a sub-class of the gliding motion class with the pili (also known as twitching) [84], which concerns individual, sparse, and sessile bacterial cells and is favored on soft surfaces [85] in a range of speed (0.06–1.4 µm s^−1^), consistent with the bacterial speeds estimated using the modeled results reported here (0.01–4 µm s^−1^; for the calculation, see Equation (7)). Twitching is usually mediated by Type-IV pili [86], which *E. coli* K12 lacks under laboratory conditions, despite the presence of the appropriate genes [87]. However, as noted by Harshey et al. [88], several different mechanisms may produce gliding in a combined manner. Some of them are probably still unknown.

Pili-related motion through attachment/detachment cycles was reported as not being free by Sjollema et al. [89]. However, this may depend on the surface properties, as the tether strengths and detachment/attachment dynamics should depend on them. In particular, the surface stiffness may be more or less favorable to detachment/attachment cycles depending on the deformation of the surface enabled at the pili-tethering locations when the attached bacterium is rolling (Figure 5C). Our hypothesis is that a softer surface reduces the traction force applied on the pili during rolling compared to that which happens on a stiffer surface. This may prevent the pili from strengthening the link to the surface, as expected in the case of a catch-bond submitted to increasing forces, or the tether from detaching if the force becomes higher than the force threshold needed to break the catch-bond. In such a scenario, an optimal stiffness should exist, depending on the rupture force of the catch-bond, which should allow the bacterium to freely move while staying attached to the surface due to the pili tethers. Such a motion should require the replacement and, thus, the production of more pili compared to situations where the bacterium remains attached to a constant location. Therefore, with this hypothesis, the optimal stiffness enabling such mobility on PDMS is here estimated to be close to a few kPa or a few tens of kPa, as PDMS-9kPa is more favorable to free motion and a higher pili abundance than PDMS-574kPa. On the HA-44Pa and HA-2kPa surfaces, such a motion may be hindered by the hydration layer that prevents the attachment of the pili to the surface. On these surfaces, as well as on PDMS-574kPa, bacterial motion may only be governed by agitation in the medium, combined with friction forces between the bacterium wall and the surface. These forces should depend on the bacterium-to-surface indentation, directly correlated with the surface stiffness, and thus be responsible for the increase in the diffusion coefficient observed when the stiffness increases and for the higher confinement observed on the very soft HA-44Pa.

## 4. Conclusions

Our results confirm that elasticity is a surface property that can be optimized to regulate the formation of biofilms on materials. However, this benefit is counterbalanced by a high degree of surface hydration. We established that increasing the softness is an efficient way of disturbing bacterial mobility and inhibiting biofilm formation on HA-based materials. Importantly, free-moving bacterial subpopulations were the rarest on the softest HA (HA-44Pa), which revealed the most confined and least diffusive subpopulations and was the least colonized material. In contrast, the biofilm formation progress occurred ahead of time on the non-hydrated PDMS (PDMS-9kPa) compared to the hydrated HA material with a similar stiffness (HA-2kPa). In line with this, a protein of Type-1 pili, which tethers bacteria to surfaces, was detected in large quantities on the PDMS-9kPa surface, while several overproduced proteins showed an early phase of bacterial adhesion. The mobility of the sessile bacteria also varied from one surface to another. In general, their diffusion coefficients increased when the elasticity increased, while their confinement was oppositely correlated with biofilm formation. This shows that freedom of mobility, but not diffusion, is favorable to biofilm formation and suggests that it could be a predictive indicator of biofilm formation. The overall findings also suggest that the mobility observed on the most colonized PDMS-9kPa material is related to its use by bacteria of Type-1 pili through a turnover of their attachment and detachment from the material surface. Finally, this study offers important insights into the methods that may be used to improve the elasticity and hydration of biomaterials formed of PDMS, HA, and other hydrogel materials so as to prevent their infection with bacteria.

## Figures and Tables

**Figure 1 jfb-13-00237-f001:**
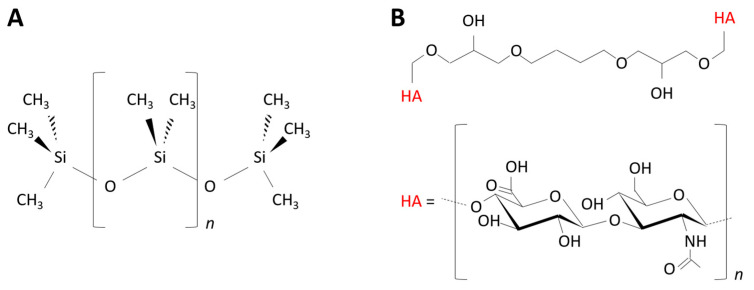
Chemical structures of the poly(dimethyl)siloxane (PDMS) (**A**) and hyaluronic acid (HA) (**B**) polymers prepared in this study.

**Figure 2 jfb-13-00237-f002:**
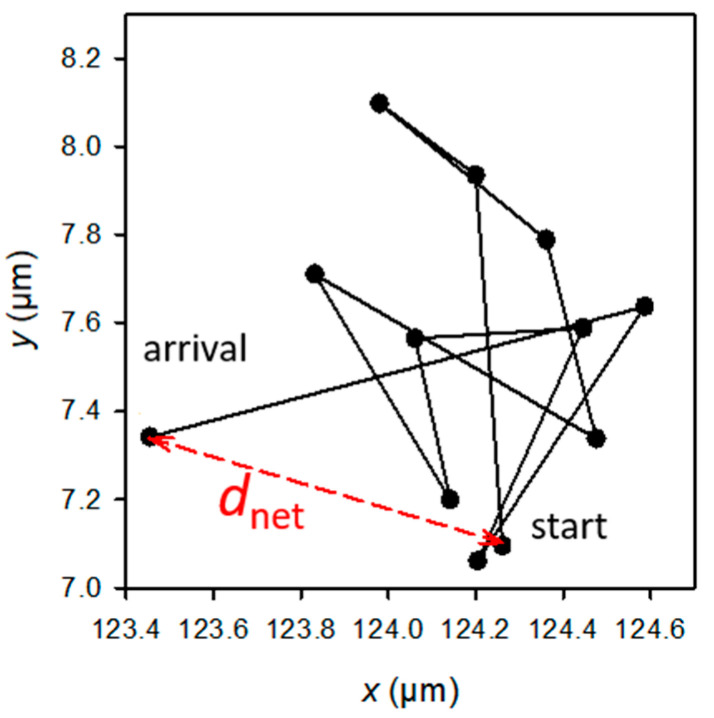
Example of the trajectory of a bacterium observed at a given *t* time of the experiment, and schematic definition of *d_net_* (in red). The sum of the black segments gives the total length, *d_tot_*, of the trajectory. Each dot corresponds to the position of the bacterium at the successive times from 0 to *t* (*t* from 0 to 55, 57, or 76 s according to the experiment).

**Figure 3 jfb-13-00237-f003:**
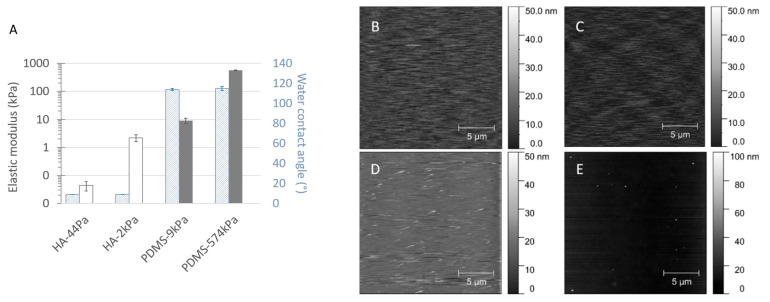
Elastic, hydrophobic, and topographic properties of the material surfaces, displayed as the Young’s moduli *E* and the water contact angle at equilibrium (**A**) and as AFM micrographs of the HA-44Pa (**B**), HA-2kPa (**C**), PDMS-9kPa (**D**) and PDMS-574kPa (**E**) materials. Mean Young’s moduli were determined by nanoindentation for each material type (average and standard deviation from 17 to 29 measurements).

**Figure 4 jfb-13-00237-f004:**
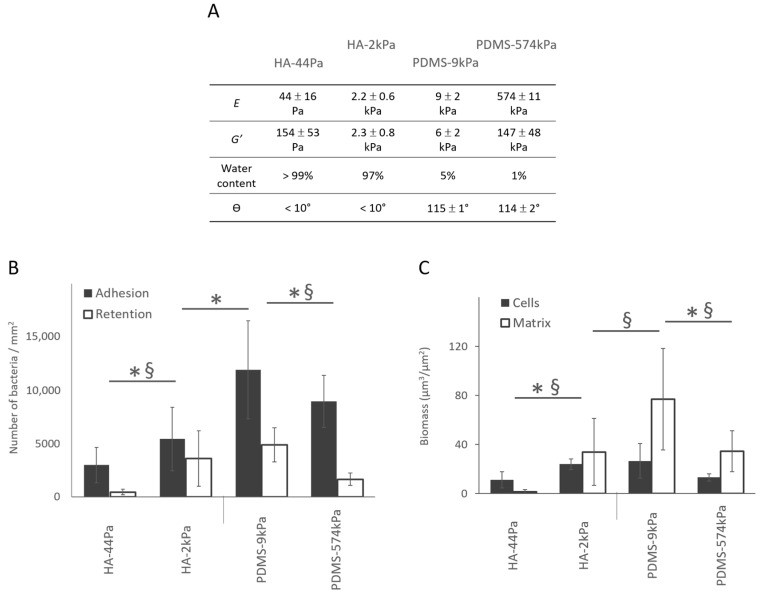
Bacterial adhesion, retention, and biofilm formation on the HA and PDMS materials. (**A**) Summary of the main properties of the HA-44Pa, HA-2kPa, PDMS-9kPa and PDMS-574kPa materials. *E*, *G*′, and θ are Young’s modulus, the elastic modulus, and contact angle, respectively. (**B**) Adhesion (gray) and retention (white) after 3 h of culture, as determined using the micrographs (Syto9^®^ staining). *, §: significant difference in adhesion and retention, respectively (*p*-value < 0.05). (**C**) Bacterial cell biomass and amount matrix production of biofilms formed after 72 h of culture, as determined using the micrographs (Syto9^®^ and Texas Red^®^ concanavalin A stains, respectively). *, §: significant difference in cell and matrix quantities, respectively (*p*-value < 0.05).

**Figure 5 jfb-13-00237-f005:**
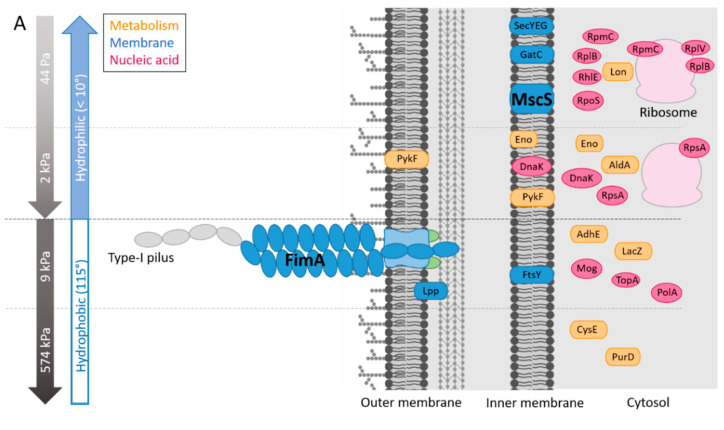
Summary of the influences of the PDMS and HA surface elasticity and surface hydration levels on (**A**) the protein abundance in the *E. coli* proteome, as determined by proteomic analyses (the highest abundances are shown), (**B**) the mobility and biofilm development of the *E. coli* sessile populations, as indicated by the integrated results, and (**C**) hypothesis of the roles of friction forces and pili tethers in the bacterial mobility on the surfaces.

**Figure 6 jfb-13-00237-f006:**
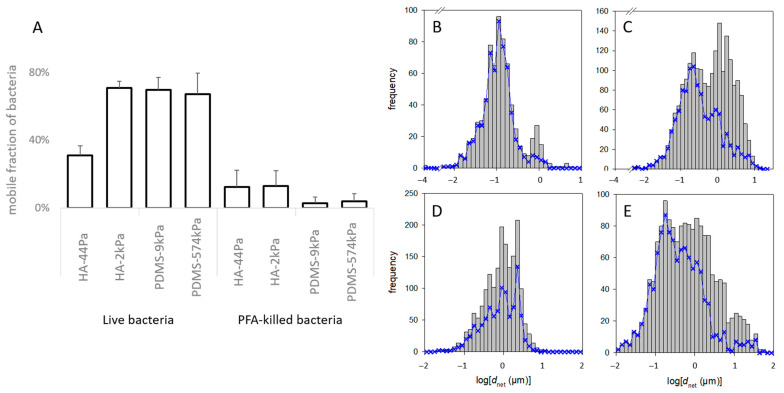
Mobility of bacterial cells on the HA and PDMS materials. (**A**) Fraction of live and dead bacteria with a displacement of more than 2 µm in the entire observation period (“mobile fraction”). (**B**–**E**) Frequency of *d_net_* of the bacteria on the HA-44Pa (**B**), HA-2kPa (**C**), PDMS-9kPa (**D**) and PDMS-574kPa (**E**) materials. The grey histograms encompass all the tracks recorded over the entire observation time (i.e., about 1 min) at the successive times when pictures were taken (e.g., 3, 6, …, 57 s). The blue curves correspond only to those tracks that could be followed over the whole observation time (e.g., 57 s).

**Figure 7 jfb-13-00237-f007:**
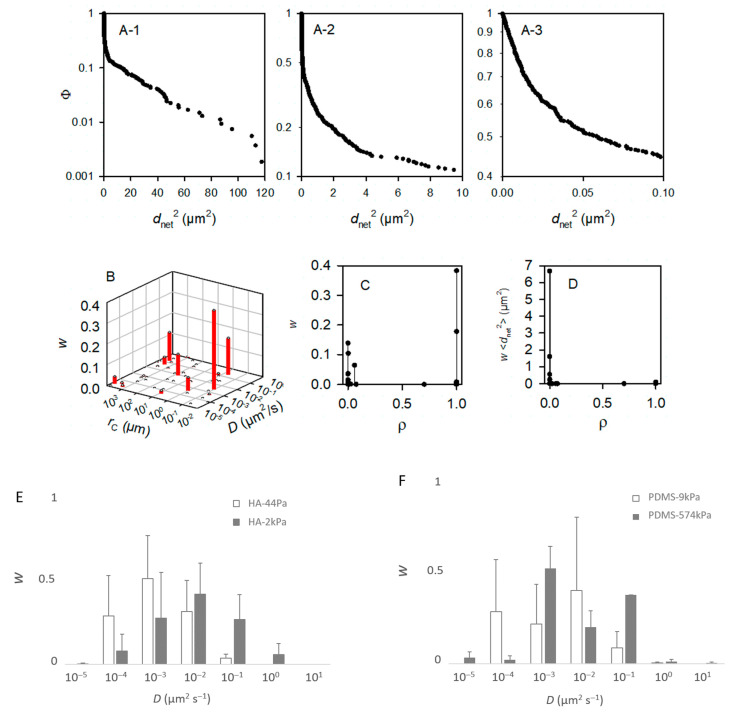
Modeling of the movement of the bacteria. From (**A**–**D**), the bacteria were deposited on a PDMS-574kPa sample, and the model accounted for *m* = 50 subpopulations. As made visible by (**B**), less than *m* = 10 subpopulations are sufficient to describe the population. Panels (**E**–**H**) were, accordingly, based on computations with *m* = 10. (**A**) Experimental complementary cumulative frequency Φ of <*d_net_*^2^> at various scales (**A-1**–**A-3**) of the *x*-axis for the purpose of showing that Φ is not a single exponentially decreasing function. (**B**) Relative weight, *w*, of the subpopulations characterized by their specific *D* and *r_C_* parameters. (**C**) Relative importance of the free (*ρ*~0) and confined (*ρ*~1) bacterial subpopulations. (**D**) Contribution of the subpopulations characterized by a confinement index *ρ* to the mean square displacement, or <*d_net_*^2^>. (**E**–**G**) Relative fractions, *w*, of the main bacterial subpopulations (∑*w_i_* > 99%) in the whole population as a function of the diffusion coefficient, *D* (*D* are grouped into [0.5 × 10^−z^, 5.0 × 10^−z^] classes with z ∈ {−1, 0, 1, 2, 3, 4, 5}): comparison between the HA-44Pa and HA-2kPa materials (**E**), between the PDMS-9kPa and PDMS-574kPa materials (**F**), and between the HA-2kPa and PDMS-9kPa materials (**G**). (**H**) Fractions, *w*, of the main bacterial subpopulations as a function of the confinement index, *ρ*.

**Figure 8 jfb-13-00237-f008:**
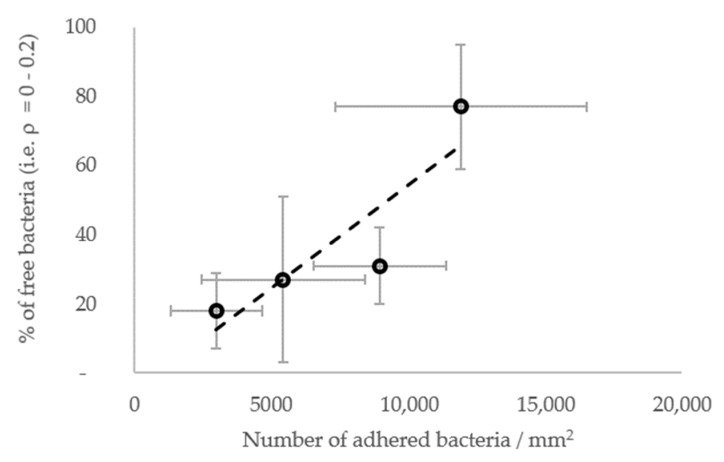
Fraction of free bacteria (i.e., with *ρ* from 0 to 0.2) depending on the resulting further bacterial adhesion. The most significant linear correlation is given by the broken line (R^2^ = 0.78).

**Table 1 jfb-13-00237-t001:** Chemical composition of the PDMS and HA material surfaces, as evaluated by high-resolution XPS analysis.

	Peak	Component	Position (eV)	Atomic %	
**PDMS**				**PDMS-9kPa**	**PDMS-574kPa**
	C1s	C–Si	284.3	44	46
	O1s	Si–O	532.0	28	27
	Si2p	Si 2p_3/2_ SiO	101.8	22	24
		Si 2p_3/2_ SiO_4_ SiO_2_	103.2	6	4
**HA**				**HA-44Pa**	**HA-2kPa**
	C1s	C–C C–H	285.0	15	25
		C–O	286.5	49	45
		C=O	288.0	3	2
	O1s	CO	532.9	34	29

**Table 2 jfb-13-00237-t002:** Differentially expressed proteins in *E. coli* sessile cells on the HA-44Pa and HA-2kPa materials. MFC: max fold change.

Function		Name	Highest Condition	MFC
**Metabolism**	Protein quality control	Lon Lon protease	HA-44Pa	19.7
	Dehydrogenase activity	AldA Lactaldehyde dehydrogenase	HA-2kPa	7.8
	Glycolytic process	Eno Enolase	HA-2kPa	5.3
	Glycolytic process	PykF Pyruvate kinase I	HA-2kPa	5.2
	Galactidol metabolic process	GatC PTS system galactitol-specific EIIC component	HA-44Pa	5.6
**Membrane**	Translocation	SecYEG Protein translocation channel SecYEG	HA-44Pa	17.1
	Mechanosensitive channel	MscS Small-conductance mechanosensitive channel	HA-44Pa	5.0
**Nucleic Acid**	DNA	DnaK Chaperone protein DnaK	HA-2kPa	17.6
	rRNA	RplV 50S ribosomal protein L22	HA-44Pa	6.3
		RpmC 50S ribosomal protein L29	HA-44Pa	5.2
		RplB 50S ribosomal protein L2	HA-44Pa	5.2
		RpoS RNA polymerase sigma factor RpoS	HA-44Pa	5.1
		RpsA 30S ribosomal protein S1	HA-2kPa	5.7
		RhlE ATP-dependent RNA helicase RhlE	HA-44Pa	6.0

**Table 3 jfb-13-00237-t003:** Differentially expressed proteins in *E. coli* sessile cells on the PDMS-9kPa and PDMS-574kPa materials. MFC: max fold change.

Function		Name	Highest Condition	MFC
**Metabolism**	Carbon utilization, Ethanol biosynthetic process	AdhE Aldehyde-alcohol dehydrogenase	PDMS-9kPa	4.0
	Cysteine biosynthesis	CysE Serine acethyltransferase	PDMS-574kPa	4.0
	Molybdenum-cofactor biosynthesis	Mog Molybdopterin adenylyltransferase	PDMS-9kPa	2.6
	Lactose catabolic process	LacZ Beta-galactosidase	PDMS-9kPa	3.2
	IMP biosynthetic process	PurD Phosphoribosylamine–glycine ligase	PDMS-574kPa	2.9
**Membrane**	Pilus organization and cell adhesion	FimA Major type-1 subunit fimbrin (pilin)	PDMS-9kPa	15.0
	Protein targeting to membrane	FtsY Signal Recognition particle receptor	PDMS-9kPa	2.7
	Lipid modification, periplasmic space organization	Lpp Major outer membrane lipoprotein	PDMS-9kPa	2.5
**Nucleic Acid**	DNA replication, damage, and repair	PolA DNA polymerase I	PDMS-9kPa	4.9
	DNA topological change	TopA DNA topoisomerase I	PDMS-9kPa	4.6

**Table 4 jfb-13-00237-t004:** Examples of proteins differently expressed on the HA and PDMS materials. (**A**) Four proteins were significantly less expressed on both HA-2kPa and PDMS-9kPa (*p*-value < 0.05) compared to HA-44Pa and PDMS-574kPa, respectively. adhE is a protein that is positively correlated with biofilm formation, whereas the other proteins could not be related to bacterial adhesion or biofilm formation. (**B**) FimA, the main sub-unit of Type-1 pili located in the outer cell membrane, was identified on the HA and PDMS materials. On the PDMS materials, the protein was in significantly higher abundance on PDMS-9kPa (MFC of 15). On the HA materials, the difference in FimA abundance cannot be considered as significant, with a *p*-value > 0.05. It should be noted that FimH was not identified on the material. This was expected, as FimH is the subunit located on the Type-1 pili tip. It could thus only be identified in the secretome [75].

(**A**)						
**Name**			**Short Name**	**MFC**		
				**PDMS**	**HA**	
NADH:ubiquinone oxidoreductase, chain G			nuoG	1.8	1.6	
pyruvate dehydrogenase, decarboxylase component E1, thiamin-binding			aceE	2.4	1.7	
fused acetaldehyde-CoA dehydrogenase/iron-dependent alcohol dehydrogenase/pyruvate-formate lyase deactivase			adhE	4.0	2.6	
phenylalanine tRNA synthetase, beta subunit			pheT	1.6	1.6	
(**B**)						
**Material**	** *p* ** **-Value (ANOVA)**	**MFC**	**Mean Abundance**			
HA	8.92 × 10^−2^	1.9	1.5 × 10^5^ (HA-2kPa)			
PDMS	2.08 × 10^−3^	15.0	1.6 × 10^6^ (PDMS-9kPa)			

## Data Availability

The mass spectrometry proteomic data were deposited in the ProteomeXchange Consortium via the PRIDE [90] partner repository with the dataset identifier PXD035233.

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
