# Peer review of "Escherichia coli Biofilm Formation, Motion and Protein Patterns on Hyaluronic Acid and Polydimethylsiloxane Depend on Surface Stiffness"

_jfb, 2022, doi:10.3390/jfb13040237_

Round 1

Reviewer 1 Report

In the paper entitled "Escherichia coli biofilm formation, motion and protein patterns on hyaluronic acid and polydimethylsiloxane depend on surface stiffness" the authors extensively investigate the effects of surface elasticity on E.coli bacteria. The experimental techniques have been chosen correctly to derive valuable conslusions. The depth of the analysis is astonishing, and the results are thoroughly described and discussed. All except one - AFM imaging. To my surprise, the description of Figure 3 B-E was few pages after the place of this figure. The Authors areasked to think about relocating the figures to the places where they will be the most informative.

In the paper "Escherichia coli biofilm formation, motion and protein patterns 2 on hyaluronic acid and polydimethylsiloxane depend on sur-3 face stiffness" the Authors are investigating an important, and usually neglected, issue which is the effect of materials elastic properties of the development of bacterial biofilms. It is an original and interesting topic, with a high impact on the design of biomaterials. It is known but usually forgettable that the term "biocompatibility", which is a general requisition for any biomaterials, refers not only to mammalian cells but also to bacteria. While designing materials facilitating cell adhesion and outgrowth, researchers usually forget that they will also facilitate the adhesion of bacteria and the development of bacterial biofilms. This paper shows that it is really the case that should be considered. The paper extensively describes the research and provides a lot of experimental data. The only limitation is that the description of specific experiments should be better organized within the text. The example is AFM data, which appear in the form of figure on page 9 but are discussed on page 12.

The conclusions are consistent with the evidence, references are appropriate.

Author Response

We thank the Reviewer 1 for his/her positive and attentive review. We moved the description of the Figure 3B-E from page 12 to page 9. The changes are highlighted in yellow. We preferred, however, to keep the discussion of this Figure on page 12, after the results on surface chemistry; this allows to discuss the effects of both material surface chemistry and topography on bacterial adhesion.

Reviewer 2 Report

The study is intensively examed on the characterization of the biofilm and its impacts HA on the biofilm. It is also included the whole proteome and other related analyses to reveal the impacts of HA on the E. coli biofilm adhesion. 

The reviewer is impressed with the study. However, it will be slightly shortened in the method, for example, the bacterial strains and growth medium can be used for other sections: bacterial culture in mobility analysis and etc...

Author Response

We are honored to hear that the reviewer found the study impressive. We attempted to shorten Materials and Methods section, which was found too long by the reviewer. We moved the description of preparation of killed bacteria to from page 5 section "Bacterial culture for mobility analysis of killed bacteria" to page 4 section "Bacterial strains and growth medium". This allowed us to eliminate some repetitive sentences from page 5. Other details, even if they may seem too long, are essential to allow reproduction of the results by other researchers, therefore we preferred to keep them unmodified.